# Urinary Incontinence Affects the Quality of Life and Increases Psychological Distress and Low Self-Esteem

**DOI:** 10.3390/healthcare11121772

**Published:** 2023-06-15

**Authors:** AlJohara M. AlQuaiz, Ambreen Kazi, Nada AlYousefi, Lemmese Alwatban, Yara AlHabib, Iqbal Turkistani

**Affiliations:** 1Princess Nora Bent Abdullah Research Chair for Women’s Health, Deanship of Scientific Research, King Saud University, Riyadh 11421, Saudi Arabia; jalquaiz@yahoo.com (A.M.A.); lalwatban@ksu.edu.sa (L.A.); or alhabib330@ksau-hs.edu.sa (Y.A.); 2Department of Family & Community Medicine, College of Medicine, King Saud University, Riyadh 11421, Saudi Arabia; 3Department of Obstetrics & Gynecology, College of Medicine, King Khalid Hospital, King Saud University Medical City, Riyadh 11461, Saudi Arabia; iqbalzmt@hotmail.com

**Keywords:** urinary incontinence, QoL, psychological distress, self-esteem, Saudi Arabia

## Abstract

Urinary incontinence is a common problem among women of reproductive age. The objectives of this study were to measure the prevalence of urinary incontinence and the association with quality of life, psychological distress and self-esteem in Saudi women in the city of Riyadh. A questionnaire-based cross-sectional study was conducted in primary healthcare centers with Saudi women aged between 30 and 75 years. The questionnaire consisted of Urinary Distress Inventory, Incontinence Impact Questionnaire, Kessler Psychological Distress Scale, Rosenberg Self-esteem Scale, and the Female Sexual Function Index. Around 47.5% of women were suffering from urinary incontinence. The most common type of incontinence was stress (79%), followed by urge (72%) and mixed type (51%). Multivariate logistic regression analysis found that stress (5.83 (3.1, 11.1)), urge (3.41 (2.0, 5.8)), mixed (8.71 (3.4, 22.4)) incontinence and severe urinary distress (8.11 (5.2, 12.7)) were associated with impaired quality of life. Women suffering from stress and urge incontinence were twice (2.0 (1.3, 2.2)) as likely of reporting moderate/severe mental distress. Women suffering from urge incontinence (1.92 (1.4, 2.7)) and severe urinary distress (1.74 (1.1, 2.8)) were at a higher prevalence of reporting low self-esteem. Urinary incontinence affects the physical, psychological, social, and sexual health of women. Healthcare providers should be knowledgeable about the adverse consequences of UI on women’s personal and social life, and provide counseling and treatment accordingly.

## 1. Introduction

Urinary Incontinence (UI) is a common, though under-reported, problem since many women refrain from seeking medical advice [1]. Although incontinence is not a life-threatening condition, it can profoundly affect the quality of life (QoL). As defined by the international continence society, UI is a “complaint of involuntary leakage of urine which is a social or hygienic problem” [2]. According to the International Urogynecological Association (IUGA) and the International Continence Society (ICS), stress urinary incontinence (SUI) is defined as urinary leakage in the presence of an increased intra-abdominal pressure without detrusor activation [2]. Urge urinary incontinence (UUI) is the involuntary loss of urine in correspondence with detrusor activation and represents a symptom of overactive bladder (OAB) [2], that is accompanied or immediately preceded by urgency [2]. Both stress and urge incontinence often coexist and are termed as mixed urinary incontinence (MUI) [2]. 

Large variations have been reported by women regarding the prevalence of UI. A recently published descriptive study from America based on the 2015–2018 National Health and Nutrition Examination Survey (NHANES) found that around 62% of women suffer from UI, with 32.4% of all women reporting symptoms on a monthly basis [3]. Of those with UI, 37.5% had stress incontinence, 22% had urge incontinence, 31.3% had mixed symptoms, and around 9% had unspecified incontinence. Around 22% were suffering from moderate to severe UI [3]. 

A literature review from China, conducted on UI and associated factors across the adult life-course of Chinese women, found that overall UI prevalence ranged from 8.7 to 69.8%. This review included 48 articles published between January 2013 and December 2019. Different age categories demonstrated differences in prevalence. For women aged 17–40 years, 41–59 years, and 60 years and older, prevalence rates ranged from 2.6 to 30.0, 8.7 to 47.7, and 16.9 to 61.6%, respectively [4]. Another systematic review was conducted focusing on epidemiology and population-based studies to identify the prevalence for various lower urinary tract symptoms (LUTS) in women. This review reported that the prevalence of LUTS in women was 11.8–88.5%. All forms of UI were observed in 5.8–45.8% of women. The majority of patients (1.9–31.8%) suffered from stress UI, followed by urge (0.7–24.4%) and mixed-type of UI (2.1–12%) [5]. 

A systematic review was conducted to review the published literature regarding the prevalence of UI, its social impact and help-seeking behavior in women belonging to the Gulf region [6]. The review study reported that the prevalence of UI ranged from 20.3% to 54.5%, and stress incontinence was the most common type [6]. Prevalence studies from Saudi Arabia conducted on the female population have shown prevalence ranging from 30% to 41.4% [7,8]. The Saudi national survey on pelvic floor dysfunctions reported that the prevalence of SUI, UUI, and MUI was 31.7%, 22.9%, and 18.4%, respectively [8]. 

Research studies have tried to identify the significant factors associated with UI. Several studies have found that the factors most strongly associated with UI were increasing age, obesity, high parity, menopause, diabetes, hypertension and smoking [3,4,5,6,7,8]. According to estimates, every additional ten years of age increases the risk of UI by 1.36-fold [4]. A systematic review and meta-analysis that included 29 studies, found the prevalence of UI in older women in the world was 37% (95% CI: 29.6–45.4%). The highest prevalence of urinary incontinence was reported in older women in Asia with 45.1% (95% CI: 36.9–53.5%) [9].

The majority of studies on UI and QoL are from Western society. The impact of UI on a woman’s life is dependent on factors such as UI subtype, presence of comorbidities, age, and socioeconomic status. From respondents to a questionnaire-based study, social functioning and general mental health were the most affected domains in all UI subtypes [10], while role limitation and emotional problems were more apparent in urge UI [10]. Physical function problems were mostly found with stress type incontinence [10]. Regarding the effect of urinary incontinence on sexual life, strong links were found between the women’s dissatisfaction with their sexual lives and having orgasmic dysfunction, and their constant concern over possible leakage during intercourse [11,12]. Another systematic review on the strength of association between UI and QI concluded that UI is strongly associated with a poor QoL [13]. However, despite the high prevalence of UI and its impact on QoL, most women show a lack of knowledge about pelvic floor muscle dysfunction, do not understand the treatment options, and are not able to identify risk factors for these disorders [14].

The systematic review from Gulf countries reported that there was a large impact on quality of life with major interference with prayers (34–90%) and sexual relationships (18–57%) [6]. The main reasons for not seeking medical advice were embarrassment to see doctors, especially male doctors, and the belief that UI is common, normal or an incurable disease [6]. A national level study on pelvic floor dysfunctions conducted on 2289 non-pregnant women from Saudi Arabia reported that 9.3% of participants use sanitary pads, and 14.5% reported an issue with prayer, as it required them to perform ablution before each of the five prayers [8].

Although several health-related reforms have been implemented, the cultural environment still does not openly discuss problems associated with women’s reproductive and sexual health [12]. Cultural differences are expected in terms of how the women perceive UI, and whether they are limited in conducting their day-to-day activities and the association with mental and sexual health [8]. Considering the cultural contrast between the Saudi female population and other global female populations, there is a need to explore the UI association with different domains of QoL. Among those are physical, mental, and social activities and emotions. The objectives of this study were to measure the prevalence of urinary incontinence, its types and the severity of UI, and to explore the association with quality of life, psychological distress, self-esteem and sexual health in Saudi women in Riyadh. 

## 2. Materials and Methods

### 2.1. Study Design, Setting, and Participants

This is a cross-sectional survey conducted from December 2015 to June 2016. This study was part of a larger research project, the “Women in Saudi Arabia Health Examination Survey”, commonly known as “WISHES”. Riyadh is the capital city of Saudi Arabia. The population of Riyadh represents different ethnicities and various socioeconomic groups. Riyadh has five administrative regions: north, south, east, west, and central. However, for sampling purpose Riyadh city was considered as a single stratum. A list of 100 primary health care centers (PHCC) was obtained from the Ministry of Health, out of which, 25 PHCCs were randomly selected using the random sampling program (accessed on 15 August 2015 on https://www.random.org/) [15]. It was assured that the selected PHCCs were from the different administrative regions of Riyadh. Some of the selected PHCCs were not accessible due to renovation work; hence, we were only able to include 18 PHCCs in total. In addition, we included private organizations (5 schools, 2 technical colleges, 4 Government offices, and 1 social welfare organization) so that working women are equally represented in our sample. 

Multiple strategies were adopted to invite and recruit the participants. These included placing 1–2 advertisement roll-ups, and informational material (25–50 one-page invitations) at each of the selected PHCCs, the largest mosque and the nearest largest shopping malls, one week before the start of the study. Invitations were disseminated to other family members through patients/attendants visiting the PHCCs. The inclusion criteria were mentioned on roll-ups, pamphlets, and letters, stating that any Saudi woman aged 30–75 years; permanent resident of Riyadh city; and who could understand and write the Arabic language were eligible to participate. Pregnant women and those mentally disabled were excluded from the study. Invitation flyers were written in the local language, stating the study’s primary objective. Neither compensation nor any type of payment was offered to participants. Around 2029 women agreed to participate and were asked to read, understand, and sign the consent form in Arabic. Out of those, 963 women had symptoms of UI for more than six months and were included in the final analysis. 

### 2.2. Data Collection Procedure

The “WISHES” study included different sections on women’s health including chronic diseases, smoking, physical activity, sun exposure, lifestyle, reproductive health, psychological distress, self-esteem, sexual health, urinary incontinence and violence against women. The researchers trained a team of five Arabic-speaking female data collectors to conduct the interviews. The training was given for one week on how to conduct the interviews, especially how to ask sensitive and personal questions. The data collectors were trained to develop rapport with the participants; be able to explain the study’s aims and significance; be able to answer their questions or queries; be able to decrease their apprehensions related to confidentiality; and make them feel comfortable in answering all type questions. In addition, training was given on how to conduct anthropometric measurements and on how to collect, store and transport blood samples. All the data collectors were asked to follow the standard protocol while filling the questionnaire, taking measurements and collecting the blood samples. For mothers with young children, a special corner was set up to engage small children in playing and coloring activities. Training sessions were followed by a pilot study of 50 participants to pretest the questionnaire and identify any logistic or technical issues. A separate room with a closed door was arranged for the interviews, each participant was assigned a unique ID, and participants’ identification details were kept confidential during data entry, analysis, and the write-up of the manuscript.

### 2.3. Data Collection Tools and Scales

As mentioned above, this study is part of a large cross-sectional survey, the “WISHES” study, and according to the objectives of this manuscript, relevant variables were included in the analysis. The Section 1 of the questionnaire included information on sociodemographic factors, including age, marital status, occupation, income, number of pregnancies, and whether they were diabetic or not. The detailed questionnaire was developed in the Arabic language and comprised previously validated questions. 

The Section 2 included the urinary distress inventory scale (UDI-6) [16]. The scale was previously validated in the Arabic language [17]. The Cronbach alpha for the UDI-6 was 0.80. UDI-6 consists of 6 items: 1. frequent urination, 2. leakage related to feelings of urgency, 3. leakage related to activity, 4. coughing, or sneezing small amounts of leakage (drops), 5. difficulty emptying the bladder, and 6. pain or discomfort in the lower abdominal or genital area. Higher scores on UDI-6 indicate a higher disability. UDI-6 scores were calculated per standard protocol (average of the total scores/25) [16]. Total scores range from 0 to 100. Continuous scores were divided into categorical variables, low distress vs. severe distress, based on a previously identified cut-off point of 33.33 [18]. 

The Section 3 included the incontinence impact questionnaire short form (IIQ-7) [16] to assess the impact on QoL. The scale has been validated in the Arabic language [19]. The Cronbach alpha for the IIQ-7 was 0.89. The data collectors administered the questionnaire, which comprised seven types of activities, as follows: daily chores (cooking, house cleaning, laundry, etc.), physical recreation (walking, swimming, or other exercises), entertainment activities (movies, concerts, etc.), ability to travel by bus or car for more than 30 min from home, social activities (get togethers, parties, etc.) and emotional health (nervousness, depression, etc.). The participants were inquired about how much the specific activity was affected due to UI. Each item was answered using the Likert scale, where a score of 0 was “not affected at all”, 1 for “affected slightly”, 2 for “moderately affected”, and 3 for “greatly affected”. All the responses were summed together, and each participant’s average score was calculated. The average ranged from 0 to 3, which was further multiplied by 33.1/3 to transfer the scores on a scale of 0 to 100. In order to convert the continuous scores to a categorical variable, they were divided based on a predetermined cut-off value of 9.52 [18]. 

The Section 4 consisted of the interviewer-administered Kessler Psychological Distress Scale [20], known as K-10. It is a global questionnaire for measuring psychological distress. The scale has been validated in Arabic and utilized by previous researchers [21]. The Cronbach alpha value for K-10 was 0.86. It consists of 10 items about anxiety and depressive symptoms over the past four weeks, each with a five-level response scale. The scores ranged from 10 to 50, with a score between 10 and 19 indicating no distress, a score between 20 and 24 indicating mild distress, a score between 25 and 29 indicating moderate distress, and a score between 30 and 50 indicating severe distress. The K-10 was converted to a categorical variable by summing together “no distress” and “mild distress” responses and coded as “0”, whereas moderate and severe distress responses were added and coded as “1”.

The Section 5 included the Rosenberg Scale for Self-Esteem (RSES) [22]. The scale has been translated and utilized in previous studies from Saudi Arabia [23]. The Cronbach alpha for the self-esteem scale was 0.75. The RSES is a self-administered global scale that measures self-confidence by measuring positive and negative emotions using ten questions, with each question following Likert responses which range from strongly agree = 3, agree = 2, disagree = 1 to strongly disagree = 0. Items 2, 5, 6, 8, and 9 were inversely coded. The responses were summed to obtain a total score ranging from 10 to 40, with high scores indicating high self-esteem. The total score variable was normally distributed; hence, we took the mean value as a cut-off. Based on the mean cut-off value of 20, scores were divided into low self-esteem (scores < 20) and high self-esteem (>20). 

The Section 6, sexual health, was assessed through the 6-item Female Sexual Function Index (FSFI-6) [24]. The short FSFI-6 is validated and available in the Arabic language [25]. The Cronbach alpha value for the FSFI was 0.76. It comprises six domains: desire, arousal, lubrication, orgasm, satisfaction, and pain. The responses were measured using a 5-point Likert scale, ranging from 1 to 5. The total scores were divided into two categories based on the mean cut-off score of 3.0. Participants scoring >3 were labeled as suffering from poor sexual health and coded as “1” and those scoring ≤3 were coded as “0”. 

### 2.4. Anthropometric Measurements

Anthropometric indices included weight, measured with an electronic scale (Secca 220—Hamburg, Germany, 2009), and height, measured following the standard protocol for a stadiometer. Weight and height were used to calculate the body mass index (BMI) as weight (kg)/(height in m^2^). BMI was divided into three categories using the internationally recommended cut-offs [26]. Normal BMI was defined as 18.5–24.9, overweight was defined as a BMI of 25.0–29.9, and obesity was defined as a BMI ≥ 30.

### 2.5. Ethical and Safety Protocol

The safety and well-being of the participants were ensured at each stage. The participants with moderate to severe urinary incontinence and K-10 scores ≥25 were asked to follow up with their family physician in the PHCC. This study was approved by the Institutional Review Board, King Saud University (E-12-658) and the Institutional Review Board of the Ministry of Health, Dammam (IRB MOH0151).

### 2.6. Statistical Methods

The data for 963 women were analyzed using the Statistical Package for Social Sciences^®^ version 21 (IBM Corp., Armonk, NY, USA). Descriptive analysis was conducted which included the frequency and percentages of the categorical variables and mean, and standard deviation for the continuous variables. Pearson correlation coefficients (r) were calculated to measure the relationship between different continuous variables. To measure the association between UI, its types and severity with QoL, psychological distress, self-esteem and sexual health, first Chi-square test was done to measure unadjusted odds ratio and 95% CI. This was followed by multivariate logistic regression analysis to measure adjusted odds ratio and 95% CI. The exposure variables were the types of incontinence and the severity of distress due to incontinence. Items under the UDI-6 scale were treated in two ways: the questions for stress, urge and mixed incontinence were taken as separate independent exposures and participants replying “greatly” in response to the question about botheration were labeled as “1”, compared to others labeled as “0”. To measure the severity of UI, total scores for UDI-6 were calculated as per the protocol and further converted to a categorical variable, with participants scoring >33.33 labeled as “1”, showing severe urinary distress, and those with scores <33.33 as “0”, denoting low distress [18]. The outcome variables were QoL, psychological distress, self-esteem and sexual health. For the outcome variable, such as IIQ, the continuous scores were converted to categorical variable according to the protocol [18]. For psychological distress, moderate and severe distress categories (K-10 categories) were added and labeled as 1, with no/mild labeled as 0. For self-esteem and sexual health, the mean scores for each scale, were taken to identify the “at risk category”, and labeled as 1, compared to those having less than the mean scores labelled as 0. Adjusted odds ratios and 95% CIs were calculated for each of the four final models with QoL (IIQ-7), psychological distress, self-esteem and sexual health showing associations with different types of UI and severities of UDI-6. All associations were adjusted for potential confounders such as age, educational level, number of pregnancies, obesity, and diabetes status. A *p* value < 0.05 was taken as significant. 

## 3. Results

### 3.1. Incontinence Prevalence and Types

The original sample of the WISHES study included 2029 women, out of which 963 (47.5%) complained of some type of urinary incontinence and were included in the analysis. Table 1 displays the sociodemographic characteristics of the participants. The mean age of the participants was 46.38 (±10.92) years, and around 80% of them were currently married. Educational level found that around 21% of women were illiterate, whereas 14% had some primary level education, 28% reported intermediate/secondary level education, and around 37% were highly educated with postgraduate qualifications. Education and age were significantly associated (*p* < 0.00) as the majority (38.4%) of the elderly women (≥60 years) were illiterate and only had a primary-level education, whereas 66% of women aged ≤44 years were diploma holders or had a postgraduate-level education. Most of the women reported a monthly household income <10,000 SAR. Education and income were significantly associated (*p* < 0.00), as the majority with high education reported >20 k SAR monthly income, whereas, those with low education level mentioned monthly income of <10 k SAR.

The occupations reported by working women were either teachers (14%), secretaries (13%), or doctors/ health staff (8%). No significant difference was observed between type of occupation and UI (*p* > 0.05). Only 2.6% of the women reported smoking. Around one-third (33%, *n* = 315) of women reported menopause. The average number of children per woman was five to six; however, 10% had more than nine children. The mean BMI was above 30 m^2^/kg, indicating obesity (32.40 (±6.63)), with only 11% of women having a BMI <25 kg/m^2^. Around 71% (*n* = 628) were diabetic, with mean HbA1c levels of 6.13 (±1.54). No significant association was found between occupation, income, and type of incontinence (*p* > 0.05).

### 3.2. Urinary Distress Inventory (UDI)

Table 2 shows the urinary distress inventory variables. The median score was 16.66, ranging from a minimum score of 4.17 to a maximum score of 66.67, and an interquartile range of 20.84. The most common type of incontinence was stress incontinence, reported by 79% (*n* = 762) of women, followed by urge incontinence (72%, *n* = 692) and mixed-type incontinence (51%, *n* = 491). The categorical cut-off 33.33 [12] found around 18% (*n* = 171) of women reporting severe urinary distress due to incontinence. Age was significantly associated with UDI scores. Univariate analysis calculated the unadjusted odds ratio and 95% CI and found that participants aged 45–59 and ≥60 years were 1.44 (1.0, 2.08) and 1.65 (1.02, 2.67) times at higher odds for severe urinary distress, respectively. Low educational level/illiteracy were partially marginally significantly associated with urinary distress (UOR 1.48 (1.0, 2.19)). No significant association was observed with marital status, income, and occupation in univariate and multivariate analyses. 

### 3.3. Incontinence Impact Questionnaire (IIQ-7)

Table 3 shows the seven items for the impact of incontinence on daily activities. The median scores for the IIQ-7 were 4.76, ranging from 0 to 99. According to the categorical cut-off of 9.52 [18], 48.7% (*n* = 469) women reported impaired quality of life (QoL) due to incontinence. The highest impairment was observed for “traveling”, as around 31% (*n* = 296) of the participants reported a moderate/severe impact on traveling using a bus/car or going for entertainment activities, followed by limited social activities (reported by 16.4%), impaired physical activity (14%) and impaired psychological health (13%). Women complaining about restricted traveling were around 50 years of age, 40% were menopausal, and around 35% suffered from diabetes mellitus. The Pearson correlation coefficient found a strong significant correlation between IIQ and urinary distress (r = 0.46, *p* < 0.00). Similarly, correlation were found between IIQ-7 scores and poor sexual health (r = 0.06, *p* = 0.04), psychological distress (r = 0.17, *p* < 0.00), and low self-esteem (r = −0.14, *p* < 0.00). In addition, IIQ-7 scores correlated significantly with increasing age (r = 0.08, *p* = 0.01) and number of children (r = 0.09, *p* = 0.008). Univariate analysis found a significant association between impaired QoL (IIQ ≥ 9.52) and urge incontinence (UOR 2.93 (2.17, 3.95)); stress incontinence (UOR 1.66 (1.21, 2.28)); and mixed incontinence (UOR 3.31 (2.54, 4.31)). 

### 3.4. Psychological Distress, Self-Esteem and Sexual Health

Table 4 displays the data regarding psychological distress. The mean scores were 22.40 (±7.67), ranging from 10 to 50. The categorical cut-off found that 25.3% had mild distress, 15% had moderate distress, and 19% reported severe distress. A majority of women with moderate/severe K-10 scores were middle-aged (mean 43 (±9.67) years), and obese with a mean BMI of 32.2 (±6.72) kg/m^2^. The most commonly reported symptom was “everything is an effort” reported by 40% of women, followed by 37% saying “feeling tired for no good reason” for most or all of the time. A Pearson correlation coefficient between continuous K-10 scores and the urinary distress inventory was significant (r = 0.137, *p* < 0.00), showing severe urinary distress is related to severe psychological distress. Similarly, a significant negative correlation was observed between K-10 scores and decreasing age (r = −0.28, *p* < 0.00). Univariate analysis found that severe/moderate mental distress was associated with a mixed type of incontinence (UOR 1.38 (1.05, 1.80)) and with severe urinary distress (UOR 1.61 (1.15, 2.25)). 

Table 5 reports the self-esteem items. The mean scores for self-esteem were 20.12 (±3.96), ranging from 8 to 30. The categorical cut-off at a mean score of 20 found that 34.6% of the participants reported low self-esteem (and were labeled as 1). The mean age of women with low self-esteem was 45.95 (±10.68) years, having a mean BMI of 32.73 (±6.71) kg/m^2^. Around one-third (31% (*n* = 105) of these women were menopausal. Self-esteem scores were not significantly associated with age or BMI. However, the Pearson correlation coefficient found that self-esteem scores were negatively correlated with urinary distress (r = −0.165, *p* = 0.00), psychological distress (r = −0.291, *p* < 0.00) and sexual health (r = −0.104, *p* = 0.004), showing women with low self-esteem had severe urinary distress, high mental distress and poor sexual health. Univariate analysis found that low self-esteem was significantly associated with severe urinary distress (UOR 1.93 (1.38, 2.69)). 

Sexual health was inquired from currently married women (*n* = 762). Keeping the cut-off value at mean score of 3.0, 17% (*n* = 131) women were found to have poor sexual health (labeled as 1). Around 30% of women complained about at least one type of sexual problem. A significant 25% (*n* = 191) of women mentioned no desire for intercourse, whereas around 16% were not satisfied after intercourse. A few (<7%) complained about dry vagina and pain during intercourse. Around 30% of women ≥60 years reported no sexual activity. Women complaining about poor sexual health were comparatively younger (mean age 42.1 (±8.65) years) with a mean BMI of 31.4 (±6.2) kg/m^2^. The Pearson correlation coefficient also found a significant negative association between sexual health scores and age (r = −0.11, *p* = 0.002); however, around 30% of women were ≥60 years and reported no sexual activity. 

### 3.5. Multivariate Logistic Regression Analysis

Table 6 shows the multivariate logistic regression table with adjusted odds ratios and 95% CI for the association between different types of incontinence and urinary distress with QoL, psychological distress, low self-esteem and sexual health. Although the percentage of women severely bothered due to stress, urge and mixed incontinence was low, 7.3% (*n* = 70), 8.4% (*n* = 81) and 4.5% (*n* = 44), respectively, it significantly affected QoL. Women with severe stress (5.83 (3.1, 11.1)), urge (3.41 (2.0, 5.8)), mixed incontinence (8.71 (3.4, 22.4)) and severe urinary distress (8.11 (5.2, 12.7)) suffered from impaired QoL. Each association was adjusted for age, education, number of pregnancies, and BMI. After adjustment, the multivariate model showed an association between UI type and severe urinary distress with IIQ-7 and found that age ≥60 years (1.57 (1.0, 2.8)); no. of pregnancies 1–4 (1.84 (1.0, 3.36)) and having DM (1.55 (1.13, 2.13)) were also significant

Women suffering from stress, urge incontinence and severe distress (based on the 33.33 cut-off) were two (or almost two) times more likely to have moderate/severe psychological distress. Similarly, women suffering from severe urge incontinence and severe distress (UDI-6 scores >33.33) were 1.92 (1.4, 2.7) and 1.74 (1.1, 2.8) times more likely to have low self-esteem, respectively. None of the different types of incontinence showed a significant association with poor sexual health; however, severe urinary distress (UDI > 33.33) was significantly associated with poor sexual health with an adjusted odds ratio of 1.91 (1.2, 3.0). Models showing an association between UI impact and its type with mental distress (using K-10) found that the number of pregnancies (1.95 (1.1, 3.8)) was significant even after adjustment. Similarly, the model showing an association between UI impact and its type with sexual health found that overweight (2.37 (1.32, 4.28)) and obesity (1.93 (1.15, 3.24)) remained significant after adjustment. However, the model showing an association between UI impact and its type with self-esteem found that none of the adjusted factors were significant. 

## 4. Discussion

UI has a high prevalence among Saudi women, which supports results from a previous review study that have also reported a high prevalence among different groups of women ranging between 25 and 45% [27]. Irvin et al. calculated the age and gender-specific prevalence for UI amongst male and females aged ≥18 years, belonging to five countries, namely, Canada, Germany, Italy, Sweden and the United Kingdom [28]. Later, using the same data, Irvin et al. projected the population estimates for UI globally and found the highest number of people suffering from UI belong to Asia, followed by Europe, Africa, North America and South America [29]. The review concludes that the differences in prevalence are attributed towards using different definitions for urine incontinence (UI) and variations in population sampling techniques; however, there is a consensus that UI is a major public health issue among women. Previous studies from Saudi Arabia have reported a prevalence between 30 and 41%, which is less than that reported by our study [7,8,30]. Except for the National study conducted by Al-Badr et al. [8], The majority of previous studies were of a small sample size, fewer number of sampling units (PHCCs), or were hospital-based studies and included non-Saudi women; hence, the generalizability of the results is limited. In addition, their main objective was just to measure the prevalence of UI in women. Amongst the types of UI, stress incontinence was the most common type (79%), followed by urge incontinence (72%) and mixed type of incontinence 51%. Similar results were reported by Al-Badr et al., who reported that stress incontinence is the most common type of UI [8]. 

The UDI-6 can estimate the distress caused by UI symptoms. About 18% of participants were found to have high urinary distress levels secondary to incontinence. All three types of UI and severe urinary distress significantly affected QoL, as the odds ratios ranged between 3.41 and 8.71. IIQ-7 is a validated tool for estimating the negative impact of UI on QoL. A study reported that almost one-half of the participants with UI reported that it negatively impacted their job and daily outdoor activities [30]. This is comparable to the current study result, which reported that also about 50% of participants had impaired QoL due to UI. The highest impairment was observed for the “travel domain”. In contrast to our results, the above study reported that the effect on the ability to travel accounted for only 16.5% of participants [30]. The difference can contribute to the variation in the study’s population. Most participants who reported travel restrictions were older, menopausal, and had diabetes.

The second most affected domain was social activities, which was reported by 16.4% of participants. The same limitations for social activities were reported by around 36% of patients in a Saudi study [31]. Similarly, UI patients in another recent Saudi study had the lowest scores in the social limitation domain, which indicate better QoL [32]. Impaired physical activity is reported by 14% of this study’s participants, which is comparable to the percentage reported by Alshammari et al.’s study (18.55) [31]. Limited travel and social activities may lead to adverse mental health effects. 

In accordance with an Iranian study [33], the current study found similar findings, i.e., women with mixed incontinence reported significantly lower QoL than those with stress and urge incontinence. Psychological distress, poor self-esteem, increasing age, having DM, and multiparity, were significantly associated with impaired QoL.

In a religious-oriented community, the impairment related to UI in mandatory ablution before prayer, having to do so five times per day, can add to the distress. Different local studies reported this. For example, 33.8% of the Al-Badr et al. study’s participants found it difficult to pray on time [8]. Several studies in Muslim countries reported an inability to pray regularly and on time, secondary to incontinence issues [31,32,33,34]. Under-reporting, and poor health-seeking behavior for assessment and management were reported in some studies [33,34]. This can add to the distress of UI patients and increase their suffering.

A meta-analysis of twelve articles found that participants with UI had significantly higher depression and anxiety levels than others [35]. We also found that more than a third of participants reported moderate/severe psychological distress. Women who were severely bothered due to stress, urge incontinence, and urinary distress on UDI-6 were at higher risk of having moderate/ severe psychological distress. The majority of them were middle-aged and obese. 

Sarikaya et al. reported a high sexual dysfunction among UI patients [36]. The severity of distress on UDI-6 can predict poor sexual health in the current study. This is in accordance with a Polish study that reported that more incontinence symptoms are associated with worse sexual satisfaction [37]. Around 30% of women complained about at least one sexual problem. A lack of desire for intercourse was reported by 25% of participants. One explanation is that urinary incontinence during intercourse may cause adverse effects on the desire and arousal stage as it can be embarrassing for both partners, especially the female, leading to the avoidance of sexual activity [37,38].

Previous studies have found that chronic diseases impact urinary incontinence. The reported prevalence of UI by the National Health and Nutrition Examination Survey (NHANES) among patients with chronic obstructive pulmonary disease was 34.9% [39]. Around 15% of women in our sample suffered from lung disease (asthma, COPD). Studies have concluded that increasing age and chronic cough are the major factors explaining the association between the two. 

We did not adjust for chronic cough; however, we adjusted for age, education, diabetes, obesity and number of pregnancies. In a population-based survey conducted in Brazil, age, diabetes, and impaired mobility were associated with UI [40]. Around 29% of the participants in this study had diabetes, and urge and mixed type of incontinence were significantly associated with having diabetes. A recent study published in 2020 that included 398 diabetic women from Taif, a city in the western province of Saudi Arabia, found that 34% of them suffer from UI (*p* < 0.05) [41]. A study from Denmark found that the prevalence of UI was 50.3% in diabetic participants and 39.3% in participants without diabetes mellitus (DM) [42]. The pathophysiology of UI in diabetes is related to diabetic autonomic neuropathy in the form of overactive bladder, voiding dysfunction, and urinary retention [43,44]. Liu et al. have shown that the prevalence of UI, especially for urge incontinence, is positively associated with the duration of DM [45]. In contrast to the above finding, data from the National Health and Nutrition Examination Survey (NHANES) led to the conclusion that after adjusting for BMI, diabetes was not independently associated with UI despite an increased prevalence of UI among women with diabetes [46]. 

The same study reported that age, parity, functional mobility limitations, and BMI were reported as risk factors for all types of UI [46]. Being overweight or obese was significantly associated with increased intra-abdominal pressures that can stress the pelvic floor muscles and may cause urinary incontinence [47], which in turn can also affect QoL. More than half of the women in our study were obese (mean BMI 32.40 (±6.63)), and the independent model showing an association between UI and sexual health found obesity to remain significant after adjusting for potential confounders. 

A study conducted to assess the impact of pregnancy on pelvic floor anatomy found that connective tissue weakness and high fetal weight at birth are important contributors to pelvic floor dysfunction [48]. In a study conducted in Dammam, a city in the eastern province of Saudi Arabia, the stress UI was statistically significantly linked to grand multiparity, i.e., five and more births [49]. Another study found that parity significantly predicts stress and urgency UI [50]. A meta-analysis of studies on UI during the postpartum period found that multiparity is a significant factor in developing UI [51]. Another meta-analysis reported that parity was associated with an increased risk of overall and stress UI, but not urgency UI [52]. The mean number of pregnancies in the current study was five to six and was correlated significantly with UI distress.

### Limitations

This study has some limitations. The data are based on patient’s self-reporting and, therefore, may be subjected to recall and misclassification types of bias. In addition, the cross-sectional nature of this study makes it difficult to establish causal relations. However, the reliability and validity of the tools used are documented in the literature, which is one of the major strengths of this study. In addition, the sample includes women belonging to different ethnicities and social classes, thus making our results generalizable.

There is a need to conduct prospective longitudinal studies that may span through the life course of women starting from menarche to old age. In addition to establishing the causal pathways, such studies can help in identifying the actual age at which the problem starts so that preventive strategies can be implemented earlier. In addition to observational studies, randomized controlled trials can help in evaluating the effectiveness of different treatment modalities or investigating interventions that can improve women’s overall health and sexual health. 

## 5. Conclusions and Recommendations

UI is common and often disturbing, especially in elderly Saudi women. Our study found that UI and its types negatively impact quality of life (QOL), increase psychological distress, and result in low self-esteem and poor sexual health. The association between UI and poor QoL may lead to a vicious circle, where poor QoL may predispose women to depression and lack of hygiene, thus further exposing the women to urinary tract infections and incontinence. Surgical treatments for the pelvic floor muscles may cause additional burdens on the family and the health care system. In addition, mothers’ poor QoL have a negative effect on the children and the family. In certain situations, marital relationships are also negatively affected.

A holistic approach is required while dealing with major health problems such as UI. Policy makers, educators, practitioners and the society at large need to participate and implement the specific recommendations. The majority of women seek health care through PHCCs. Policy makers should ensure that “Well Women Clinics” are available at all the government-run facilities to provide screening and treatment for health issues specifically associated with women. In addition, clean and maintained toilet facilities should be available in all public places to avoid any inconvenience or embarrassment to women. 

Health educators can play an important role by introducing the subject through social media and TV talk shows to create awareness about UI and its associated impact on the women’s QoL. Cognitive behavioral therapy, when implemented, has a positive effect on patients’ mental health and QoL; hence, the health educators through “Well Women Clinics” can arrange group counseling sessions to overcome the anxiety associated with UI and improve mental health through relaxation techniques. In addition, public campaigns can be arranged to raise awareness by distributing educational material and to recognize UI as a public health problem. This will help women initiate conversations about this topic without any hesitancy and will encourage women to seek treatment.

Healthcare providers should be aware of the negative consequences of UI on women’s physical, social, and mental health. It is overdue that, similar to other chronic diseases, UI should be treated as a priority problem. An initial screening checklist can help in identifying the patients at an early stage, thus avoiding an increase in severity and future complications. In addition, a physiotherapist can educate women on pelvic floor exercises to strengthen the muscles and avoid further complications. 

## Figures and Tables

**Table 1 healthcare-11-01772-t001:** Distribution of sociodemographic and health-related variables in women with urinary incontinence in Riyadh, Saudi Arabia (*N* = 963).

Variables	Frequency (%)
Age (in years)	
30–44	432 (44.9)
45–59	390 (40.5)
≥60	141 (14.6)
Total	963 (100)
Marital status	
Married (currently)	762 (79.1)
Widow	117 (12.1)
Divorced/separation	84 (8.8)
Total	963 (100)
Participants Education	
Illiterate	198 (20.6)
Primary	135 (13.9)
Intermediate and Secondary	270 (28.1)
≥Graduate	360 (37.4)
Total	963 (100)
Participants occupation	
Housewife	547 (56.8)
Working	383 (39.8)
Retired	33 (3.4)
Total	963 (100)
Income (in SAR)	
>20,000	84 (8.7)
10,000–20,000	261 (27.1)
<10,000	618 (64.2)
Total	963 (100)
Number of pregnancies	
None	59 (6.1)
1–4	272 (28.2)
>4	632 (65.6)
Total	963 (100)
Menopause status	
No	648 (67.3)
Yes	315 (32.7)
Total	963 (100)
Health Indicators	
Body Mass Index (kg/m^2^)	
Normal	104 (10.8)
Over weight	244 (25.3)
Obese	615 (63.9)
Total	963 (100)
Diabetes Mellitus	
No	682 (70.8)
Yes	281 (29.2)
Total	963 (100)

**Table 2 healthcare-11-01772-t002:** Urinary Distress Inventory variables reported by women with urinary incontinence in Riyadh, Saudi Arabia.

Variables	Frequency (%)
Urinary Incontinence ^1^ (*n* = 2029)	
No	1066 (52.5)
Yes	963 (47.5)
Type of Incontinence (*n* = 963)	
Stress ^2^	762 (79.1)
Urge ^3^	692 (71.9)
Mixed ^4^	491 (51)
Do you suffer from frequent urination?	
No	384 (39.9)
Yes	579 (60.1)
Do you experience difficulty in emptying the bladder ^5^	
No	748 (77.7)
Yes	215 (22.3)
Pain in the lower abdomen or genital area (pelvic region) in context to urination	
No	523 (54.3)
Yes	440 (45.7)
Urinary distress inventory categories	
Distressed (<33.33)	792 (82.2)
Severely distressed (≥33.33)	171 (17.8)

^1^ Women who complained about involuntary leakage of urine. ^2^ Stress urinary incontinence was defined as the complaint of involuntary leakage on effort or exertion, or on sneezing or coughing. ^3^ Urge urinary incontinence was defined as the complaint of involuntary leakage accompanied by or immediately preceded by urgency. ^4^ Mixed urinary incontinence was defined as the complaint of involuntary leakage associated with urgency and also with exertion, effort, sneezing, or coughing. ^5^ Hesitancy refers to difficulty in emptying the bladder.

**Table 3 healthcare-11-01772-t003:** Incontinence Impact Questionnaire-7 (IIQ-7) reported by women suffering from urinary incontinence in Riyadh, Saudi Arabia (*N* = 963).

Has Urine Leakage Affected Your Ability to Do	Not at All	Slightly	Moderately	Greatly
Household chores	715 (74.2)	155 (16.1)	63 (6.5)	30 (3.1)
Physical recreation	669 (695)	198 (20.6)	61 (6.3)	35 (3.6)
Entertainment activities	655 (68)	197 (20.5)	79 (8.2)	32 (3.3)
Drive for 30 min by car or bus	514 (53.4)	166 (17.2)	153 (15.9)	130 (13.5)
Socialize outside the home	589 (61.2)	216 (22.4)	113 (11.7)	45 (4.7)
Emotional health	659 (68.4)	196 (20.4)	72 (7.5)	36 (3.7)
Feeling frustrated	747 (77.6)	165 (17.1)	40 (4.2)	11 (1.1)
Incontinence Impact Questionnaire				
No/low impact (<9.52)	494 (51.3)			
Impaired (≥9.52)	469 (48.7)			

**Table 4 healthcare-11-01772-t004:** Psychological distress reported by women suffering from urinary incontinence in Riyadh, Saudi Arabia (*N* = 963).

K-10 Items	Most/All the Time *n* (%)	Little/Not at All *n* (%)
Feel tired for no good reason	359 (37.3)	604 (62.7)
Feeling nervous	296 (30.7)	667 (69.3)
So nervous that nothing can calm you down	125 (13)	838 (87)
Feeling hopeless	76 (7.9)	887 (92.1)
Feeling restless and fidgety	138 (14.3)	825 (85.4)
So restless, you cannot stand it	177 (18.4)	786 (81.6)
Feeling depressed	124 (12.9)	839 (87.1)
Everything (any activity) is an effort	385 (40)	578 (60)
Nothing can cheer you up	105 (11)	858 (89)
Feeling worthless	57 (5.9)	906 (94.1)
Psychological distress (K-10)		
No/Mild (scores <20)	634 (65.8)	
Moderate/ Severe (scores ≥20)	329 (34.2)	

**Table 5 healthcare-11-01772-t005:** Self-Esteem items reported by women suffering from incontinence in Riyadh, Saudi Arabia (*N* = 963).

Rosenberg Self-Esteem Scale	Agree/Strongly Agree *n* (%)	Disagree/Strongly Disagree *n* (%)
1. On the whole I am satisfied with myself	877 (91)	86 (9)
2. At times, I think I am no good at all	231 (24)	732 (76)
3. I feel that I have a number of good qualities	937 (97)	27 (3)
4. I am able to do things as well as most other people	917 (95)	46 (5)
5. I feel I do not have much to be proud of	327 (34)	636 (66)
6. I certainly feel useless at times	157 (16.3)	806 (83.7)
7. I feel that I’m a person of worth, at least on an equal plane with others	906 (94)	57 (6)
8. I wish I could have more respect for myself	24 (25.2)	720 (74.8)
9. All in all, I am inclined to feel that I am a failure	118 (12.2)	845 (87.8)
10. I take a positive attitude toward myself	857 (90)	106 (10)
Self-Esteem		
High Self-Esteem (>20)	630 (65.4)	
Low Self-Esteem (≤20)	333 (34.6)	

**Table 6 healthcare-11-01772-t006:** Multivariate logistic regression analysis showing the adjusted odds ratio and 95% CI for the association of urinary incontinence with QoL, psychological distress, self-esteem and sexual health in women in Riyadh, Saudi Arabia.

Variables	Stress Incontinence		Urge Incontinence		Mixed-Type Incontinence		Total Urinary Distress Inventory Scores ^5^	
	Severe Distress *n* = 70	Adjusted Odds Ratio (95%CI)	Severe Distress *n* = 81	Adjusted Odds Ratio (95%CI)	Severe Distress *n* = 44	Adjusted Odds Ratio (95%CI)	Severe Distress *n* = 171	Adjusted Odds Ratio (95%CI)
**IIQ-7** ^1^								
Low Impact	12 (17.1)	**1.0**	20 (24.7)	**1.0**	5 (11.4)	**1.0**	25 (14.6)	**1.0**
Impaired	58 (82.9)	**5.83 (3.1, 11.1)**	61 (75.3)	**3.41 (2.0, 5.8)**	39 (88.6)	**8.71 (3.4, 22.4)**	146 (85.4)	**8.11 (5.2, 12.7)**
**K-10** ^2^								
No/Mild	35 (50)	**1.0**	43 (53.1)	**1.0**	24 (54.5)	1.0	97 (56.7)	**1.0**
Moderate/severe	35 (50)	**2.01 (1.2, 3.3)**	38 (46.9)	**1.99 (1.2, 3.2)**	20 (45.5)	1.68 (0.90, 3.2)	74 (43.3)	**1.78 (1.3, 2.5)**
**Self-Esteem** ^3^								
High	41 (58.6)	1.0	43 (53.1)	**1.0**	24 (54.5)	1.0	90 (52.6)	**1.0**
Low	29 (41.4)	1.33 (0.81, 2.2)	38 (46.9)	**1.74 (1.1, 2.8)**	20 (45.5)	1.57 (0.85, 2.9)	81 (47.4)	**1.93 (1.4, 2.7)**
**Sexual health** ^4^								
Satisfactory	28 (50.9)	1.0	33 (49.3)	1.0	14 (42.4)	1.0	65 (48.1)	**1.0**
Poor	27 (49.1)	1.32 (0.76, 2.3)	34 (50.7)	1.37(0.83, 2.25)	19 (57.6)	1.92 (0.94, 3.9)	70 (51.9)	**1.61 (1.1, 2.4)**

^1^ The model showing an association between UI and IIQ-7 found that age ≥60 years (1.57 (1.0, 2.8)); no. of pregnancies 1–4 (1.84 (1.0, 3.36)) and having DM (1.55 (1.13, 2.13)) were also significant. ^2^ The model showing an association between UI and K-10 found that the no. of pregnancies (1–4) was (1.95 (1.1, 3.8)) significant. ^3^ The model showing an association between UI and self-esteem found that none of the adjusted factors were significant. ^4^ Inquired from married mothers only. The model showing an association between UI and sexual health found that overweight (2.37 (1.32, 4.28)) and obesity (1.93 (1.15, 3.24)) were significant. ^5^ UDI-6 was categorized into severe and less severe based on the cut-off of 33.33. Values in bold signify *p* < 0.05.

## Data Availability

Data can be made available on special request to the principal investigator.

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
