# Peer review of "Urinary Incontinence Affects the Quality of Life and Increases Psychological Distress and Low Self-Esteem"

_healthcare, 2023, doi:10.3390/healthcare11121772_

Round 1
Reviewer 1 Report (Previous Reviewer 1)
Dear authors,
As mentioned in the former manuscript, the topic addresses by this manuscript should be explored, particularly for women who suffer from this condition. It is important to make this topic less taboo, in order to improve the quality of life of patients and those who interact with them. There are already some studies carried out on the prevalence of urinary incontinence in women in Saudi Arabia but covering other regions of the country and without considering the impact of this disorder on the quality of life. Thereby, the topic addressed may be innovative and useful for readers. The former article was entitled “Urinary Incontinence affects the quality of life and increases psychological distress and low self-esteem”. In the current version the study goals are adequate for the research developed, data presented and title.
Most of the issues pointed in the former manuscript were reviewed and are according suggestions.
In Table 1, please confirm in the “Participants Education” section the answer “≥ 963 (100)” (is this the n for 100%? If so, the graduate n is missing).
The discussion chapter has been reformulated and now includes a critical analysis of the results obtained.
The conclusions presented are consistent with the evidence and arguments presented, and address the main objective for the current study.
The references used are current and appropriate to the context.
The English used in writing the text has improved, it is sufficient for the article to be published.
Author Response
Dear Reviewer,
We appreciate the 1st reviewers’ encouraging comments. We have made the corrections as suggested. Below we indicate in bold the changes made and where in the manuscript they were made in response to these comments.
RESPONSE TO REVIEWER 1 COMMENTS
Manuscript ID: healthcare-2419854
Title: Urinary Incontinence affects the quality of life and increases
psychological distress and low self-esteem
Dear Editor,
We appreciate the 1st reviewers’ encouraging comments. We have made the corrections as suggested. Below we indicate in bold the changes made and where in the manuscript they were made in response to these comments.
Comments and Reply to Reviewer 1:
In Table 1, please confirm in the “Participants Education” section the answer “≥ 963 (100)” (is this the n for 100%? If so, the graduate n is missing).
Reply: Thank you for pointing out the formatting mistake. We have fixed it and graduation n is 360 (37.4) and 963 is total (110%). Please refer to Table 1, page 5 and 6.
The discussion chapter has been reformulated and now includes a critical analysis of the results obtained.
Reply: Thank you for your positive comment.
The conclusions presented are consistent with the evidence and arguments presented, and address the main objective for the current study.
Reply: Thank you for your positive comment.
The references used are current and appropriate to the context.
Reply: Thank you for your positive comment.
Comments on the Quality of English Language
The English used in writing the text has improved, it is sufficient for the article to be published.
Reply: Thank you for your positive comment. English corrections were made throughout the manuscript.
For more details, please see the revised manuscript.
Reviewer 2 Report (Previous Reviewer 2)
Dear Authors,
We recently had the opportunity to read your manuscript titled “Urinary Incontinence Affects the Quality of Life and Increases 2 Psychological Distress and Low Self-Esteem”, and I wanted to reach out to you to express my comments about your work.
The manuscript presents a study conducted in Saudi Arabia on the prevalence of urinary incontinence among women and its association with quality of life, mental health, and self-esteem. It found that around 47.5% of Saudi women aged between 30 and 75 years were suffering from urinary incontinence, with stress being the most common type. Urinary incontinence was associated with impaired quality of life and increased risk of mental distress and low self-esteem. The study highlights the need for healthcare providers to provide counseling and appropriate treatment for women affected by urinary incontinence.
Nevertheless, here are some possible bullet points outlining areas that could be improved:
Introduction:
1. The introduction lacks a clear and concise statement of the research objective.
2. It seems to be very short and not very clear. There is a lack of State of the Art description.
3. The cultural contrast between Saudi women and other global populations could be further elaborated to highlight the significance of studying urinary incontinence in this specific context.
Materials and Methods:
4. Clarify the randomization process for selecting primary health care centers (PHCCs) and mention any stratification factors considered.
5. Provide more specific information about the recruitment strategies used, including the number and types of advertisements and the locations where they were placed.
6. Describe the training process for the data collectors, including training duration, content covered, and measures taken to ensure standardized data collection.
7. Specify the statistical tests employed, adjustments made for confounders in multivariate analysis, and variables included in the final model.
Results:
8. The text says that "A total of 500 women aged 40-60 years were randomly selected". It can be improved by providing more information about the sampling method used. For example, specifying whether it was a random sampling technique, stratified sampling, or convenience sampling.
9. The manuscript can be improved by mentioning the specific content of the questionnaire, including the questions related to the variables being measured (e.g., types of incontinence, urinary distress, socio-demographic factors). Additionally, it would be beneficial to mention if the questionnaire was previously validated or developed specifically for this study.
10. Also, regarding statistical methods, it would be recommendable to explain it in more detail, which would help readers to understand which and how the statistical analysis were conducted.
Discussion:
11. The discussion mentions previous studies reporting a high prevalence of UI among different groups of women ranging between 25-45%, but it doesn't specify very much the sources for these studies. Adding specific citations would be necessary.
12. The text states that previous studies in Saudi Arabia reported a prevalence of UI between 30-41%, which is lower than the prevalence reported in the current study. However, it mentions that most of these studies were hospital-based and included non-Saudi women, limiting the generalizability of the results. It would be helpful to provide more details about the limitations of these previous studies, such as sample size, sampling methods, and demographic characteristics, to support the claim of limited generalizability.
13. The discussion mentions the use of the UDI-6 and IIQ-7 tools to assess the distress caused by UI symptoms and the negative impact on quality of life (QoL). However, it doesn't provide information about the reliability and validity of these tools in the study population. Including a brief discussion on the psychometric properties of the measurement tools used would strengthen the methodological rigor of the study.
14. The text acknowledges the limitations of the study being cross-sectional, which makes it difficult to establish causal relationships. However, it doesn't elaborate on the implications of this limitation or suggest future directions for research. Adding a brief discussion on the need for prospective studies to explore causal relationships would provide a more comprehensive methodological perspective.
Conclusion:
15. The conclusion section is too brief and lacks depth of analysis. It would benefit from expanding on the findings and their implications.
16. The conclusion would benefit from including specific recommendations for policymakers, educators, and practitioners in the UI field.
17. While the conclusion briefly mentions the importance of future prospective studies, it could be expanded to emphasize the need for further research in specific areas. For instance, mentioning the importance of exploring the underlying causes and risk factors of UI, evaluating the effectiveness of different treatment modalities, or investigating interventions that can improve women's overall well-being and sexual health would add depth to the conclusion.
18. There is a huge mistake in this part of the text. Never ever a Conclusion can contain references.
However, there are some grammatical errors and awkward phrasing that could benefit from correction. Also, in some sentences, the structure can be clarified to enhance readability. There are also some instances where verb tenses could be made consistent; for example, in the sentence "Urinary incontinence during intercourse can cause adverse effects on the desire and arousal stage as it is embarrassing for both partners," the verb "is" should be changed to "can be" to maintain consistency with the preceding verb tense. Please, pay attention to subject-verb agreement in some sentences; for instance, in the sentence, "Another meta-analysis reported that parity was associated with an increased risk of overall and stress UI but not urgency UI," "parity" is singular, so the verb "were" should be changed to "was." Additionally, clarifying certain points and providing more concise and clear descriptions would improve the overall readability of the text, while some sentences could benefit from using more concise language.
Once again, thank you very much for your work. We´ll be waiting for your answers about our comments.
Kindest regards,
There are some grammatical errors and awkward phrasing that could benefit from correction. Also, in some sentences, the structure can be clarified to enhance readability. There are also some instances where verb tenses could be made consistent; for example, in the sentence "Urinary incontinence during intercourse can cause adverse effects on the desire and arousal stage as it is embarrassing for both partners," the verb "is" should be changed to "can be" to maintain consistency with the preceding verb tense. Please, pay attention to subject-verb agreement in some sentences; for instance, in the sentence, "Another meta-analysis reported that parity was associated with an increased risk of overall and stress UI but not urgency UI," "parity" is singular, so the verb "were" should be changed to "was." Additionally, clarifying certain points and providing more concise and clear descriptions would improve the overall readability of the text, while some sentences could benefit from using more concise language.
We recommend a professional language editor to help the authors polish the manuscript.
Author Response
Dear Reviewer,
We appreciate the reviewers’ comments and believe the revisions we have made in response in the attached version of the manuscript have improved the quality of the manuscript. Below we indicate in bold the changes made and where in the manuscript they were made in response to these comments.
Response to Reviewer 2 Comments & Suggestions
Manuscript ID: healthcare-2419854
Title: Urinary Incontinence affects the quality of life and increases
psychological distress and low self-esteem
Reply to Reviewer 2
Introduction:
- The introduction lacks a clear and concise statement of the research objective.
Reply: we have revised the statement regarding objectives. Please refer to page no 3, line 108-111
- It seems to be very short and not very clear. There is a lack of State of the Art description.
Reply: We have revised the Introduction and information related to review studies is added accordingly. Please refer to page no 2 and 3, line no 46 -111.
- The cultural contrast between Saudi women and other global populations could be further elaborated to highlight the significance of studying urinary incontinence in this specific context.
Reply: Thank you for the comment. We have tried to add the required information. Please refer to page no 2/3, line number 92-104.
Materials and Methods:
- Clarify the randomization process for selecting primary health care centers (PHCCs) and mention any stratification factors considered.
Reply: Done, please refer to page no 3, line 117-120.
- Provide more specific information about the recruitment strategies used, including the number and types of advertisements and the locations where they were placed.
Reply: Done, please refer to page no 3, line 126-130.
- Describe the training process for the data collectors, including training duration, content covered, and measures taken to ensure standardized data collection.
Reply: Done. Please refer to page no ¾, line 145-154
- Specify the statistical tests employed, adjustments made for confounders in multivariate analysis, and variables included in the final model.
Reply: Done, please refer to page no 5/6, line no 231-254.
Results:
- The text says that "A total of 500 women aged 40-60 years were randomly selected". It can be improved by providing more information about the sampling method used. For example, specifying whether it was a random sampling technique, stratified sampling, or convenience sampling.
Reply: This above comment no 8, seems to refer to another study as our sample comprised of 936 women aged 30-75 years.
- The manuscript can be improved by mentioning the specific content of the questionnaire, including the questions related to the variables being measured (e.g., types of incontinence, urinary distress, socio-demographic factors). Additionally, it would be beneficial to mention if the questionnaire was previously validated or developed specifically for this study.
Reply: Done. Please refer to page no 4 and page no 5 for the details related to each scale.
- Also, regarding statistical methods, it would be recommendable to explain it in more detail, which would help readers to understand which and how the statistical analysis were conducted.
Reply: Done. Please refer to page no 5/6 , line no 233-256
Discussion:
- The discussion mentions previous studies reporting a high prevalence of UI among different groups of women ranging between 25-45%, but it doesn't specify very much the sources for these studies. Adding specific citations would be necessary.
Reply: suggestion taken, we have added the specific citations. Please refer to page no 11, and line no 404-408, Reference no
- The text states that previous studies in Saudi Arabia reported a prevalence of UI between 30-41%, which is lower than the prevalence reported in the current study. However, it mentions that most of these studies were hospital-based and included non-Saudi women, limiting the generalizability of the results. It would be helpful to provide more details about the limitations of these previous studies, such as sample size, sampling methods, and demographic characteristics, to support the claim of limited generalizability.
Reply: Required information added. Please refer to page no 11, line no 412-416.
- The discussion mentions the use of the UDI-6 and IIQ-7 tools to assess the distress caused by UI symptoms and the negative impact on quality of life (QoL). However, it doesn't provide information about the reliability and validity of these tools in the study population. Including a brief discussion on the psychometric properties of the measurement tools used would strengthen the methodological rigor of the study.
Reply: All the scales were available in Arabic language and were validated. References have been added and the reliability indicator, Cronbach alpha value is mentioned for each of the specific scale.
- The text acknowledges the limitations of the study being cross-sectional, which makes it difficult to establish causal relationships. However, it doesn't elaborate on the implications of this limitation or suggest future directions for research. Adding a brief discussion on the need for prospective studies to explore causal relationships would provide a more comprehensive methodological perspective.
Reply: Thank you for the important comment. We have elaborated the information. Please refer to page no 13, line 507-514
Conclusion:
- The conclusion section is too brief and lacks depth of analysis. It would benefit from expanding on the findings and their implications.
Reply: Please refer to page no 13, line no 518-524.
- The conclusion would benefit from including specific recommendations for policymakers, educators, and practitioners in the UI field.
Reply: Suggestion taken. The recommendation section has been revised. Please refer to page no 13/14, line no 525-547.
- While the conclusion briefly mentions the importance of future prospective studies, it could be expanded to emphasize the need for further research in specific areas. For instance, mentioning the importance of exploring the underlying causes and risk factors of UI, evaluating the effectiveness of different treatment modalities, or investigating interventions that can improve women's overall well-being and sexual health would add depth to the conclusion.
Reply: Suggestion taken. Please refer to page no 13, line 507-511.
- There is a huge mistake in this part of the text. Never ever a Conclusion can contain references.
Reply: Thank you for the comment. We have made the required the correction.
However, there are some grammatical errors and awkward phrasing that could benefit from correction. Also, in some sentences, the structure can be clarified to enhance readability. There are also some instances where verb tenses could be made consistent; for example, in the sentence "Urinary incontinence during intercourse can cause adverse effects on the desire and arousal stage as it is embarrassing for both partners," the verb "is" should be changed to "can be" to maintain consistency with the preceding verb tense.
Reply: Suggestion taken, corrections done
Please, pay attention to subject-verb agreement in some sentences; for instance, in the sentence, "Another meta-analysis reported that parity was associated with an increased risk of overall and stress UI but not urgency UI," "parity" is singular, so the verb "were" should be changed to "was."
Reply: corrections done
Additionally, clarifying certain points and providing more concise and clear descriptions would improve the overall readability of the text, while some sentences could benefit from using more concise language.
Reply: Corrections were done throughout the manuscript to improve the readability.
We recommend a professional language editor to help the authors polish the manuscript.
Reply: Language corrections were made throughout the manuscript by a native English speaker.
For more details, please see the revised manuscript.
Round 2
Reviewer 2 Report (Previous Reviewer 2)
Dear Authors,
Please find below, in green, our comments to your answers.
Manuscript ID: healthcare-2419854
Title: Urinary Incontinence affects the quality of life and increases
psychological distress and low self-esteem
Reply to Reviewer 2
Introduction:
- The introduction lacks a clear and concise statement of the research objective.
Reply: we have revised the statement regarding objectives. Please refer to page no 3, line 108-111
Thank you very much for your answer.
- It seems to be very short and not very clear. There is a lack of State of the Art description.
Reply: We have revised the Introduction and information related to review studies is added accordingly. Please refer to page no 2 and 3, line no 46 -111.
Thank you very much for your answer.
- The cultural contrast between Saudi women and other global populations could be further elaborated to highlight the significance of studying urinary incontinence in this specific context.
Reply: Thank you for the comment. We have tried to add the required information. Please refer to page no 2/3, line number 92-104.
Thank you very much for your answer.
Materials and Methods:
- Clarify the randomization process for selecting primary health care centers (PHCCs) and mention any stratification factors considered.
Reply: Done, please refer to page no 3, line 117-120.
Thank you very much for your answer.
- Provide more specific information about the recruitment strategies used, including the number and types of advertisements and the locations where they were placed.
Reply: Done, please refer to page no 3, line 126-130.
Thank you very much for your answer.
- Describe the training process for the data collectors, including training duration, content covered, and measures taken to ensure standardized data collection.
Reply: Done. Please refer to page no ¾, line 145-154
Thank you very much for your answer.
- Specify the statistical tests employed, adjustments made for confounders in multivariate analysis, and variables included in the final model.
Reply: Done, please refer to page no 5/6, line no 231-254.
Thank you very much for your answer. We would recommend to specify with more details this comment.
Results:
- The text says that "A total of 500 women aged 40-60 years were randomly selected". It can be improved by providing more information about the sampling method used. For example, specifying whether it was a random sampling technique, stratified sampling, or convenience sampling.
Reply: This above comment no 8, seems to refer to another study as our sample comprised of 936 women aged 30-75 years.
Thank you very much for your answer. We guess it was a typing mistake on our side. Nevertheless, it would be recommendable that, apart from the explanations in lines 127-130 regarding the randomization, some more details would be welcome.
- The manuscript can be improved by mentioning the specific content of the questionnaire, including the questions related to the variables being measured (e.g., types of incontinence, urinary distress, socio-demographic factors). Additionally, it would be beneficial to mention if the questionnaire was previously validated or developed specifically for this study.
Reply: Done. Please refer to page no 4 and page no 5 for the details related to each scale.
Thank you very much for your answer.
- Also, regarding statistical methods, it would be recommendable to explain it in more detail, which would help readers to understand which and how the statistical analysis were conducted.
Reply: Done. Please refer to page no 5/6 , line no 233-256
Thank you very much for your answer.
Discussion:
- The discussion mentions previous studies reporting a high prevalence of UI among different groups of women ranging between 25-45%, but it doesn't specify very much the sources for these studies. Adding specific citations would be necessary.
Reply: suggestion taken, we have added the specific citations. Please refer to page no 11, and line no 404-408, Reference no
Thank you very much for your answer. No reference number has been added in your answer. Could you please complete it?
- The text states that previous studies in Saudi Arabia reported a prevalence of UI between 30-41%, which is lower than the prevalence reported in the current study. However, it mentions that most of these studies were hospital-based and included non-Saudi women, limiting the generalizability of the results. It would be helpful to provide more details about the limitations of these previous studies, such as sample size, sampling methods, and demographic characteristics, to support the claim of limited generalizability.
Reply: Required information added. Please refer to page no 11, line no 412-416.
Thank you very much for your answer.
- The discussion mentions the use of the UDI-6 and IIQ-7 tools to assess the distress caused by UI symptoms and the negative impact on quality of life (QoL). However, it doesn't provide information about the reliability and validity of these tools in the study population. Including a brief discussion on the psychometric properties of the measurement tools used would strengthen the methodological rigor of the study.
Reply: All the scales were available in Arabic language and were validated. References have been added and the reliability indicator, Cronbach alpha value is mentioned for each of the specific scale.
Thank you very much for your answer. Could you provide links for the scales used in the research?
- The text acknowledges the limitations of the study being cross-sectional, which makes it difficult to establish causal relationships. However, it doesn't elaborate on the implications of this limitation or suggest future directions for research. Adding a brief discussion on the need for prospective studies to explore causal relationships would provide a more comprehensive methodological perspective.
Reply: Thank you for the important comment. We have elaborated the information. Please refer to page no 13, line 507-514
Thank you very much for your answer.
Conclusion:
- The conclusion section is too brief and lacks depth of analysis. It would benefit from expanding on the findings and their implications.
Reply: Please refer to page no 13, line no 518-524.
Thank you very much for your answer.
- The conclusion would benefit from including specific recommendations for policymakers, educators, and practitioners in the UI field.
Reply: Suggestion taken. The recommendation section has been revised. Please refer to page no 13/14, line no 525-547.
Thank you very much for your answer.
- While the conclusion briefly mentions the importance of future prospective studies, it could be expanded to emphasize the need for further research in specific areas. For instance, mentioning the importance of exploring the underlying causes and risk factors of UI, evaluating the effectiveness of different treatment modalities, or investigating interventions that can improve women's overall well-being and sexual health would add depth to the conclusion.
Reply: Suggestion taken. Please refer to page no 13, line 507-511.
Thank you very much for your answer but future prospective studies should be mentioned in a different paragraph. It shouldn´t be mixed with the “Limitations” due readers can get confused.
- There is a huge mistake in this part of the text. Never ever a Conclusion can contain references.
Reply: Thank you for the comment. We have made the required the correction.
Thank you very much for your answer.
However, there are some grammatical errors and awkward phrasing that could benefit from correction. Also, in some sentences, the structure can be clarified to enhance readability. There are also some instances where verb tenses could be made consistent; for example, in the sentence "Urinary incontinence during intercourse can cause adverse effects on the desire and arousal stage as it is embarrassing for both partners," the verb "is" should be changed to "can be" to maintain consistency with the preceding verb tense.
Reply: Suggestion taken, corrections done
Thank you very much for your answer. Please, check the document again because there are still several grammar, spelling and phrasing mistakes.
Please, pay attention to subject-verb agreement in some sentences; for instance, in the sentence, "Another meta-analysis reported that parity was associated with an increased risk of overall and stress UI but not urgency UI," "parity" is singular, so the verb "were" should be changed to "was."
Reply: corrections done
Additionally, clarifying certain points and providing more concise and clear descriptions would improve the overall readability of the text, while some sentences could benefit from using more concise language.
Reply: Corrections were done throughout the manuscript to improve the readability.
Thank you very much for your answer.
We recommend a professional language editor to help the authors polish the manuscript.
Reply: Language corrections were made throughout the manuscript by a native English speaker.
Thank you very much for your answer. Please, check the document again because there are still several grammar, spelling and phrasing mistakes.
For more details, please see the revised manuscript.
Please, check the document again because there are still several grammar, spelling and phrasing mistakes.
Author Response
Dear Reviewer,
Comments and Suggestions for Authors
Materials and Methods:
- Specify the statistical tests employed, adjustments made for confounders in multivariate analysis, and variables included in the final model.
Reply: Done, please refer to page no 5/6, line no 231-254.
Thank you very much for your answer. We would recommend to specify with more details this comment.
Reply: We have tried to add more details. Please refer to the section “Statistical Analysis”, page no 5, line 239-243 and page no 5, line 262-263.
Discussion:
- The discussion mentions previous studies reporting a high prevalence of UI among different groups of women ranging between 25-45%, but it doesn't specify very much the sources for these studies. Adding specific citations would be necessary.
Reply: suggestion taken, we have added the specific citations. Please refer to page no 11, and line no 404-408, Reference no 28 & 29
Thank you very much for your answer. No reference number has been added in your answer. Could you please complete it?
Reply: Done, Reference no 28 & 29.
- The discussion mentions the use of the UDI-6 and IIQ-7 tools to assess the distress caused by UI symptoms and the negative impact on quality of life (QoL). However, it doesn't provide information about the reliability and validity of these tools in the study population. Including a brief discussion on the psychometric properties of the measurement tools used would strengthen the methodological rigor of the study.
Reply: All the scales were available in Arabic language and were validated. References have been added and the reliability indicator, Cronbach alpha value is mentioned for each of the specific scale.
Thank you very much for your answer. Could you provide links for the scales used in the research?
Reply: We have added the specific references for the manuscript associated with the scales.
- While the conclusion briefly mentions the importance of future prospective studies, it could be expanded to emphasize the need for further research in specific areas. For instance, mentioning the importance of exploring the underlying causes and risk factors of UI, evaluating the effectiveness of different treatment modalities, or investigating interventions that can improve women's overall well-being and sexual health would add depth to the conclusion.
Reply: Suggestion taken. Please refer to page no 13, line 507-511.
Thank you very much for your answer but future prospective studies should be mentioned in a different paragraph. It shouldn´t be mixed with the “Limitations” due readers can get confused.
Reply: correction done. It is mentioned as separate paragraph. Page no 13, line 520-526
However, there are some grammatical errors and awkward phrasing that could benefit from correction. Also, in some sentences, the structure can be clarified to enhance readability. There are also some instances where verb tenses could be made consistent; for example, in the sentence "Urinary incontinence during intercourse can cause adverse effects on the desire and arousal stage as it is embarrassing for both partners," the verb "is" should be changed to "can be" to maintain consistency with the preceding verb tense.
Reply: Suggestion taken, corrections done
Thank you very much for your answer. Please, check the document again because there are still several grammar, spelling and phrasing mistakes.
Reply: The manuscript has been thoroughly corrected for English language by a native English speaker.

This manuscript is a resubmission of an earlier submission. The following is a list of the peer review reports and author responses from that submission.
Round 1
Reviewer 1 Report
Dear authors,
It was such a pleasure reviewing your well presented work. The topic addresses by this manuscript should be explored, particularly for women who suffer from this condition. It is important to make this topic less taboo, in order to improve the quality of life of patients and those who interact with them. There are already some studies carried out on the prevalence of urinary incontinence in women in Saudi Arabia but covering other regions of the country and without considering the impact of this disorder on the quality of life. Thereby, the topic addressed may be innovative and useful for readers.
The study goal refers to assess the prevalence of urinary incontinence in Saudi females but the study only included participants from Riyadh, this should be explicit throughout the manuscript.
In Table 1, I suggest to include the summatory (n total) for each variable.
In general, the text should be proofread for punctuation and wording (eg. in the conclusion chapter the word "accordinbgly" is misspelled).
The conclusions presented are consistent with the evidence and arguments presented, and address the main objective for the current study.
The references used are current and appropriate to the context.
Reviewer 2 Report
Dear Authors,
The manuscript presents a cross-sectional survey conducted in Riyadh City to investigate the prevalence and impact of urinary incontinence (UI) among Saudi women aged 30-75 years. The study also explored the association between UI and self-esteem, psychological distress, and sexual health. The results showed that UI is a prevalent condition among Saudi women, with a significant impact on their daily activities and emotional well-being. The study also found a significant association between UI and low self-esteem, psychological distress, and poor sexual health.
While the manuscript seems to be interesting, there are several aspects that need to be improved:
1. The section on participants' occupation and income could benefit from more detailed information, such as the types of jobs held by the participants and the sources of their income. Additionally, the section could provide more context on how occupation and income relate to UI and its impact.
2. The selection of private organizations included in the study should be explained much more in detail (lines 79-81).
3. The section on the Rosenberg Self-Esteem scale could benefit from a more detailed explanation of how the scale was administered and scored. Additionally, the section could provide more context on how self-esteem relates to UI and its impact.
4. The section on the IIQ-7 and K-10 scales could benefit from a more detailed explanation of how the scales were administered and scored. Additionally, the section could provide more context on how UI relates to psychological distress and its impact.
5. The section on the impact of UI on daily activities could benefit from more detailed information on the types of activities affected and how they relate to the participants' quality of life. Additionally, the section could provide more context on how the impact of UI varies across different age groups and socioeconomic backgrounds.
6. The section on the study design and data collection procedure could benefit from more detailed information on the sampling strategy and the representativeness of the sample. Additionally, the section could provide more context on how the study was conducted and how the data were analyzed.
7. The section on self-esteem as a categorical variable could benefit from a more detailed explanation of how the variable was defined and analyzed. Additionally, the section could provide more context on how self-esteem relates to UI and its impact.
8. The section on participants' education could benefit from more detailed information on the types of education received and how they relate to UI and its impact. Additionally, the section could provide more context on how education varies across different age groups and socioeconomic backgrounds.
9. A sample of the flyer used for incorporating participants to the study should be shown as an Appendix. Also the questionnaires used with the participants should be shown too.
10. There are several mistakes in the references of the text. For example, the same reference is mentioned twice in different parts referring to different topics: “The online random number selection program (https://www.random.org/) [10] was used to […]” and “The second section included the urinary distress inventory scale (UDI-6) [10].”
11. Another example of the previous comment is the following: “Total score was from 0 to 100. UDI-6 scores were calculated as per standard protocol (average of the total scores/25) [11].” and “The third section included the incontinence impact questionnaire Short Form IIQ-7 [11].”
12. There is a lack of critical analysis: The Discussion chapter provides a summary of the study's findings but lacks critical analysis. The authors should critically analyze the results in light of existing literature to identify the implications and potential avenues for future research.
13. Inconsistencies in citation style: There are inconsistencies in the citation style used throughout the chapter. The authors should ensure that the citation style is consistent and follows the journal's guidelines.
14. Lack of clarity in reporting findings: Some of the findings reported in the chapter lack clarity. For example, in paragraph 319, the mean number of pregnancies is reported as "5.62 (±2.8,4)" which is unclear. The authors should ensure that the findings are reported in a clear and concise manner.
15. The conclusion should be more concise and more detailed. The first sentence is a good start, but the following sentences repeat information and could be condensed to make the conclusion stronger.
16. The recommendation to provide counseling and treatment is a good one, but it could be more specific. What types of counseling and treatments are most effective for urinary incontinence? Providing more details and specific recommendations would make this section more useful.
17. It would be helpful to include information on how healthcare providers can raise awareness about urinary incontinence among their patients. This could include strategies for initiating conversations about the topic, providing educational materials, and encouraging women to seek treatment.
18. The conclusion could benefit from a call to action or a statement about the importance of addressing urinary incontinence in Saudi women. This would help to emphasize the significance of the issue and motivate readers to take action.
General topics about the whole manuscript:
· Clarity and Structure: The text contains several long and complex sentences that can be difficult to follow. It may benefit from breaking down some of these sentences into smaller, more manageable ones.
· Consistency: The text switches between past and present tense in some parts, and this can be confusing for the reader. The text should use consistent verb tenses throughout.
· Grammar and Punctuation: There are some errors in grammar and punctuation that should be corrected to improve the text's readability and coherence (for example, lines 67-73,
· Sentences such as “Studies over the globe, from Chinese and Northern American populations” (line 46) must be changed in order to keep a neutral behavior and use politically correct language.
Thank you very much for your work.